# Advanced Equipment Development and Clinical Application in Neurorehabilitation for Spinal Cord Injury: Historical Perspectives and Future Directions

**Yuji Kasukawa [1],\*, Yoichi Shimada [2], Daisuke Kudo [1], Kimio Saito [1], Ryota Kimura [1] , Satoaki Chida [3], Kazutoshi Hatakeyama [3] and Naohisa Miyakoshi [1]**

[1] Department of Orthopedic Surgery, Akita University Graduate School of Medicine, 1-1-1 Hondo, Akita 010-8543, Japan; dkudo@doc.med.akita-u.ac.jp (D.K.); kimio@doc.med.akita-u.ac.jp (K.S.); rkimura@med.akita-u.ac.jp (R.K.); miyakosh@doc.med.akita-u.ac.jp (N.M.)

[2] Independent Administrative Institution, Akita Prefectural Development and Disability Organization, 1-1-2 Minamigaoka, Akita 010-1409, Japan; yshimada@med.akita-u.ac.jp

[3] Division of Rehabilitation Medicine, Akita University General Hospital, 1-1-1 Hondo, Akita 010-8543, Japan; satoaki@hos.akita-u.ac.jp (S.C.); hata@hos.akita-u.ac.jp (K.H.)

\* Correspondence: kasukawa@doc.med.akita-u.ac.jp; Tel.: +81-18-884-6148

**Abstract:** Partial to complete paralysis following spinal cord injury (SCI) causes deterioration in health and has severe effects on the ability to perform activities of daily living. Following the discovery of neural plasticity, neurorehabilitation therapies have emerged that aim to reconstruct the motor circuit of the damaged spinal cord. Functional electrical stimulation (FES) has been incorporated into devices that reconstruct purposeful motions in the upper and lower limbs, the most recent of which do not require percutaneous electrode placement surgery and thus enable early rehabilitation after injury. FES-based devices have shown promising results for improving upper limb movement, including gripping and finger function, and for lower limb function such as the ability to stand and walk. FES has also been employed in hybrid cycling and rowing to increase total body fitness. Training using rehabilitation robots is advantageous in terms of consistency of quality and quantity of movements and is particularly applicable to walking training. Initiation of motor reconstruction at the early stage following SCI is likely to advance rapidly in the future, with the combined use of technologies such as regenerative medicine, brain machine interfaces, and rehabilitation robots with FES showing great promise.

**Keywords:** spinal cord injury; neural plasticity; functional electrical stimulation; rowing; cycling; rehabilitation robot

## 1. Introduction

Spinal cord injury (SCI) causes quadriplegia or paraplegia depending on the height of the injured spinal cord, and ranges from complete paralysis to incomplete paralysis depending on the degree and severity of the injury. SCI restricts activities of daily living (ADL), and severe quadriplegia makes it impossible to perform almost all ADL, requiring a great deal of care and support. The incidence and causes of SCI vary according to geographical region, social background, and culture [1,2]. Its incidence has been reported as 3.6 per million in Canada [3] and 195.4 per million in Ireland [4]. The most common causes of SCI have been reported as traffic accidents (67%) in West Africa, falls (60%) in South Asia, violence or self-harm (43%) in South Africa, and occupational accidents (23%) in Western Europe [1].

In recent years, the number of SCIs caused by falls of the elderly has increased in developed countries due to the aging of the population [5]. The latest national epidemiological survey of Japan, in 2018, which is the most aged country in the world, revealed that the proportion of elderly people is increasing, with an incidence of 49 per million and an average

age of 66.5 years with a peak in those in their 70s [6]. In Japan, the most common severity of SCI in 2018 was Frankel grade D (46.3%), and fall was the most common cause (38.6%) [6]. In addition, the rate of cervical SCI increased from 75.0% in a national epidemiological survey conducted about 30 years ago to 88.1% in 2018 [6]. When elderly people are injured, the disability becomes severe due to complications and/or comorbidities, even in mild cases. It has been reported that the physical function and ADL of older patients with SCI do not improve significantly [7]. People with SCI often spend their life in a wheelchair as the paralysis becomes more severe, and suffer numerous problems such as decreased systemic tolerability and complications of metabolic syndrome. The high morbidity and mortality rates in persons with long-term SCI stem from cardiometabolic causes, which are likely to be associated with major changes in body composition [8]. In a recent report, one-month mortality risk was significantly higher at older than 75 years compared to younger than 55 years in the patients with traumatic SCI [9].

Therefore, it is very important to improve the function of people with SCI even in elderly people. Although rehabilitation treatment is performed, mainly by physical therapy and occupational therapy, it has been difficult to restore the function of an impaired spinal cord. These conventional rehabilitation treatments also limit the recovery of injured spinal cord function because of the previous thinking that nerves do not regenerate. However, since the discovery of neural plasticity [10], the frequency and task-specificity of rehabilitation treatment and neuroplasticity have been considered very important factors for effective rehabilitation treatment [11]. In recent years, neurorehabilitation treatment has been attracting attention as a means to overcome the limits of the past and as a more effective rehabilitation treatment. Neurorehabilitation is a new treatment method that attempts to reconstruct the motor circuit of the damaged spinal cord, and is expected to be more effective than the natural recovery of the cord. Neurorehabilitation for SCI employs various devices, including functional electrical stimulation (FES) [12,13], rehabilitation robots [14], and brain-computer interfaces (BCI) [15,16].

In addition, the realization of spinal cord regenerative medicine has been promoted worldwide [17,18]. In Japan, intravenous administration of human autologous bone marrow-derived mesenchymal stem cells was performed in 13 subacute SCI patients, which resulted in improved neurological function in 12 patients [19]. In this way, new treatment methods such as stem cell transplantation have been aiming to restore neurological function, which should be used in combination with rehabilitation treatment (especially neurorehabilitation) to achieve the important therapeutic goal of enabling SCI patients to acquire independent ADL [20].

Here, we review the history of the clinical applications of the advanced devices of FES and robot rehabilitation, discuss the results of our research on neurorehabilitation for SCI, and describe the future prospects for these technologies.

The content of this paper is presented as follows:

1. FES for SCI
2. FES for upper limb paralysis

    2.2. FES for lower limb paralysis
    2.3. FES rowing and cycling

3. Robotic rehabilitation with FES for SCI
4. Future directions of FES
5. Conclusion

## 2. FES for SCI

FES is a treatment method that uses electrical stimulation to contract each paralyzed muscle and reconstruct purposeful movements for upper and lower limb motor functions impaired by central nervous system injuries such as stroke or SCI. Although paralysis occurs in SCI due to damage to the upper motor neurons, normal electrical excitement remains in the lower motor neurons and their controlling muscles. Thus, electrical stimulation is

used in place of excitatory impulses from the upper motor neurons, which can be applied directly to the lower motor neurons to contract paralytic muscles (Figure 1).

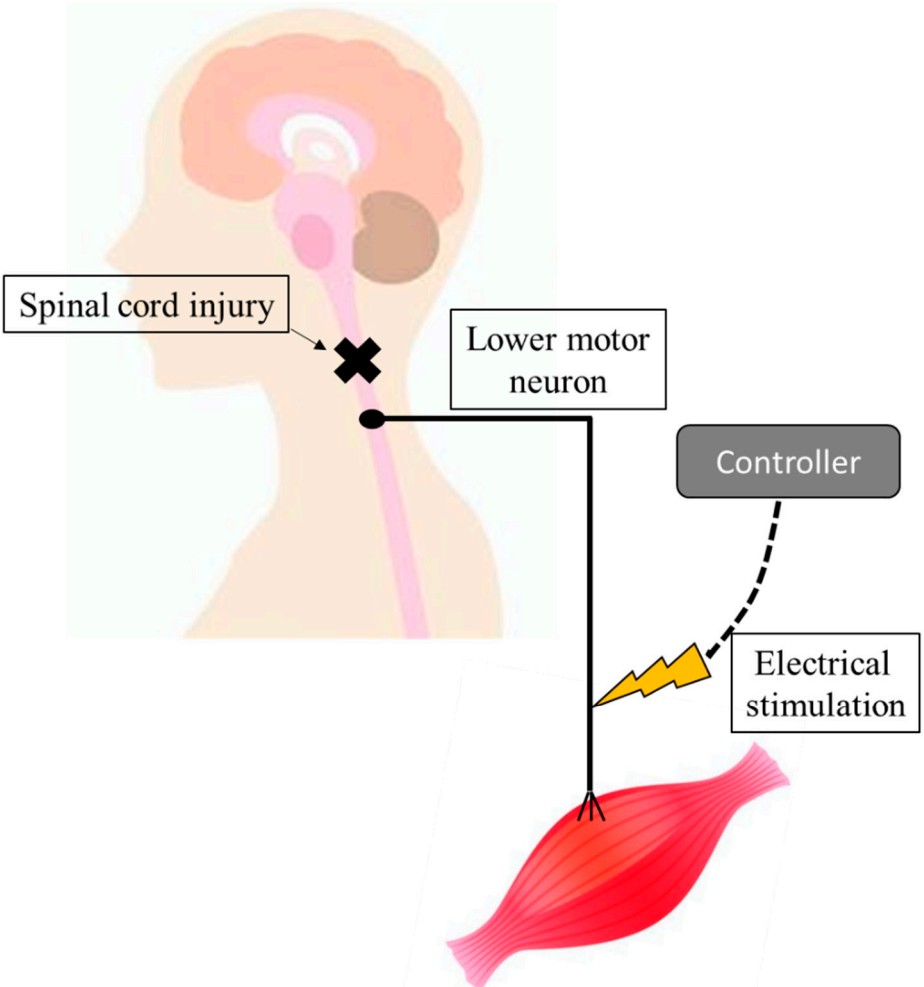

**Figure 1.** Schematic diagram of the principle of functional electrical stimulation (FES). Instead of upper motor neuron damaged by spinal cord injury, muscles contract and reconstruct function through lower motor neurons that are not damaged by controlled electrical stimulation.

In 1961, Liberson et al. reported the first use of FES, in which ankle dorsiflexion was controlled according to the gait cycle by stimulating the common peroneal nerve with a surface electrode in a patient with varus equinus due to stroke [21]. FES was first applied for SCI in 1963 by Long et al. [22], who stimulated the extensor digitorum muscle of a patient with cervical SCI to open the hand, in combination with a finger hinge splint to reconstruct ADL [22]. Subsequent studies have reported the application of FES in programmed motion stimuli to the paralyzed limb via multiple stimulus electrodes to reconstruct purposeful motions in the upper limbs for activities such as writing, eating, and drinking; and in the lower limbs for walking and standing [23]. Following these studies, randomized controlled trials were performed to investigate the effectiveness of FES training on the independence or function of SCI patients. Popovic et al. performed a randomized controlled trial to examine the efficacy of 40 h of FES therapy with conventional occupational therapy (COT) compared with COT alone [24]. The FES therapy significantly reduced disability and improved voluntary grasping in the subjects with tetraplegia. Kapadia et al. reported that 16 weeks of thrice-weekly FES-assisted walking program compared to a non-FES exercise program in chronic incomplete traumatic SCI patients [25]. The recent systematic review and meta-analysis have indicated that FES although significantly increased upper extremity

independence, there was no obvious difference in overall upper extremity function, lower extremity independence, and life quality of individuals with SCI [26].

### 2.1. FES for Upper Limb Paralysis

Following the first report of FES for upper limb paralysis in quadriplegia due to cervical SCI [20], Peckham et al. developed the FES-based system, comprising an 8-channel fully implanted electrode, a stimulator, and an external controller, and tested the system in practical use [27]. In Japan, Handa et al. developed a percutaneous implantable electrode FES system that achieved excellent control of quadriplegic upper limbs [28]. Since the 1990s, our group has been working on the reconstruction of the finger gripping function of quadriplegic upper limbs using this system, which has shown success in the reconstruction of practical movements such as eating, drinking, and writing [29].

However, as these systems require percutaneous implantable electrode placement surgery, it is difficult to start treatment early after injury or onset. Accordingly, a new surface electrode type FES device has been developed that enables the initiation of treatment from an early stage after SCI without the need for percutaneous implantable electrode placement surgery [30].

The NESS H200® Wireless system (Bioventus LLC, Durham, NC, USA) for the upper limbs is designed specifically to reconstruct finger function in paralyzed upper limbs, and consists of an orthotic component that houses a stimulation device, an internal surface electrode that attaches to the upper limb, and a control unit with a built-in FES program. The stimulation conditions can be set and transferred to the control unit by wireless operation. This device is particularly indicated for reconstruction of function in quadriplegia due to cervical SCI with residual functional levels of C5 or C6. Alon et al. validated its safety and efficacy in 5 cases of C5 paralysis and 2 cases of C6 paralysis more than 3 years after SCI [30], and reported improved upper limb function and ADL according to the Fugl-Meyer Assessment [30]. We examined the therapeutic effect of NESS H200® from an average of 6 days after paralysis in 23 cases of incomplete limb paralysis due to cervical SCI. The average treatment period was 60 days, and upper limb movement was assessed by the Japanese Orthopedic Association Cervical Myelopathy Treatment Criteria (JOA Score). Functional items (scored out of 4 points) were significantly improved from $1.7 \pm 0.5$ (mean $\pm$ standard deviation [SD]) points at the start to $2.9 \pm 0.9$ (mean $\pm$ SD) points at the end of treatment [31]. In addition, two randomized controlled trials that compared therapy with FES alone or in combination with conventional therapy reported that FES improved voluntary grasping [24,32,33]. A systematic review concluded that FES alone or in combination with conventional therapy improved the function of the upper limbs in patients with SCI [33].

### 2.2. FES for Lower Limb Paralysis

In upper limbs with quadriplegia, finger function by FES is emphasized. In the reconstruction of the function of paraplegic lower limbs; however, it is important to obtain muscular strength against gravity. Based on previous studies, it is possible to reconstruct standing and walking functions by FES for paraplegic lower limbs. However, in complete paraplegia, it is possible to walk indoors using assistive equipment such as a walker, but it is not yet practical to do so outdoors because of the risk of falling. In the case of incomplete paraplegia, practical walking indoors and outdoors is possible depending on the case.

Andrews et al. demonstrated the effectiveness of FES using surface electrodes to reconstruct lower-limb function using surface electrodes in patients with paraplegia [34], and Kraji et al. succeeded in controlling gait in completely paraplegic patients using surface electrodes [35,36]. Klose et al. developed a 6-channel surface electrode-type lower limb FES system termed the Parastep 1 Ambulation system, and reported a mean continuous walking distance of 334 m and a maximum of 1707 m in 16 patients with complete paraplegia [37]. Marsolais et al. developed the VA-CWRU (Cleveland Veterans Administration Medical Center and Case Western Reserve University) FES System, which controls movements

involved in standing, walking, and sitting using an extracorporeal stimulator with a 48-channel percutaneous implantable electrode. This system could reconstruct walking on flat ground as well as also stair climbing in paraplegics [38].

In the 1990s, our group developed an FES stimulator (Akita Stimulator I, Akita, Japan) using 18-channel percutaneous implantable electrodes, and began reconstruction of paraplegic standing and walking. We then developed the 32-channel Akita Stimulator II. This device can also apply high-frequency therapeutic electrical stimulation (TES) function for standing motion and walking swing. In addition, the standing and walking function was reconstructed by hybrid FES using short- and long-leg orthotics [39].

Similar to that in use for the upper limbs, an orthotic-type FES device was developed for rehabilitation treatment of the lower limbs from an early stage after injury. The NESS L300TM is indicated for gait disturbance due to foot drop following an upper motor neuron disease or injury including SCI. It comprises a functional stimulation (FS) cuff with an RF Stim Unit that is attached to the proximal part of the lower leg for stimulation, a control unit that sets the stimulation mode, and an Intelli-sense gait sensorTM that is attached to the foot. Smith et al. reported a case of paraparesis in which NESS L300TM was used on both sides and enabled independent walking [40]. In addition, NESS L300TM Plus, which can be stimulated simultaneously with the thigh, can control the knee joint in addition to the ankle joint [41], and is expected to expand the indications for cases of SCI.

## 2.3. FES Rowing and Cycling

Hettinga et al. reported the benefits of fitness exercise for preventing obesity, heart disease, and diabetes in people with SCI, who have a 3–5 times higher prevalence of diabetes and a 60% greater incidence of heart disease compared with healthy people [42]. Therefore, fitness exercise to prevent the deterioration of tolerability is an important approach in the rehabilitation of paraplegics. Most conventional fitness exercises for paraplegics are generally performed using the remaining voluntary function of the upper limbs. However, exercising only the upper limbs cannot prevent the progression of muscular atrophy and improve circulatory disorders in paralyzed muscles of the lower limbs. Therefore, FES has been developed in combination with such as rowing and cycling equipment to promote whole-body exercise.

### 2.3.1. FES Rowing

Laskin et al. reported the combination of rowing with FES as a hybrid exercise for people with SCI in 1993 [43]. The hybrid FES rowing system itself was developed by Wheeler and Andrews and co-workers, who reported that using the RowStim II system significantly increased rowing distance, peak oxygen consumption, and peak oxygen pulse in people with SCI [44,45]. The hybrid system was developed to enable participation in a major international rowing event in 2004 [46].

Our group also developed an FES rowing machine for preventing lower limb muscle atrophy and improving overall fitness for paraplegic patients after SCI [47]. Using this machine, the upper limbs perform voluntary movements whereas the paralyzed lower limbs perform rowing movements via FES, which are controlled by a switch attached to the handle. In paraplegic patients, FES stimulation can be performed in smooth synchronization with the rowing motion under an appropriate load [47] (Figure 2). The rowing motion exercises both the upper and lower limbs, as a whole-body exercise; in contrast, cycling exercises only the lower limbs. The FES-rowing exercise was significantly more effective at increasing peak oxygen consumption during exercise compared with arm-only exercise [44,48,49]. A recent systematic review found that the FES rowing system was a viable exercise system for individuals with SCI that can improve cardiovascular performance and reduce bone density loss [50].

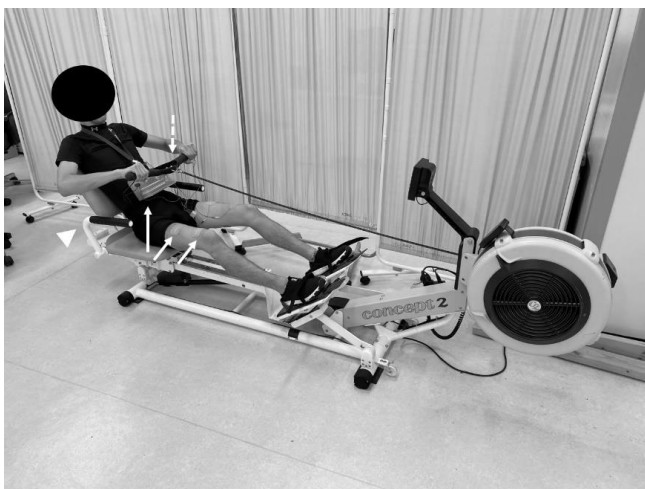

**Figure 2.** Functional electrical stimulation (FES) rowing machine. A backrest is fixed on the seat (white arrow head). A switch on the handle bar (white dotted arrow) is installed to control the FES (white arrows) timing by the patients themselves.

### 2.3.2. FES Cycling

Among the various FES training methods, FES cycling is the most commonly available and has been studied extensively [51]. Numerous studies since the 1980s have been performed regarding FES cycling after SCI. A recent systematic review of health and fitness-related outcomes of FES cycling exercise after SCI demonstrated that FES cycling improves lower-body muscle health and increases power output and aerobic fitness [52]. FES cycling is also effective in suppressing spasticity [53] and improves measures of bone health such as bone mineral density and bone metabolic markers [52]. In contrast to FES rowing, FES cycling has the advantage that it can be performed in home-based environments [54].

Our group has also developed a new FES cycling system that attaches to a standard wheelchair. It consists of a front-wheel-drive unit that attaches to the wheelchair and an FES control unit that enables cycling movements in those with lower-limb paralysis. FES stimulation output is performed by a controller that coordinates the contraction of the quadriceps femoris and hamstring with the paralyzed muscles of the lower limbs from the crank angle during cycling (Figure 3). We are currently verifying the effect of the system on lower limb muscle and bone health in patients with paraplegia due to SCI.

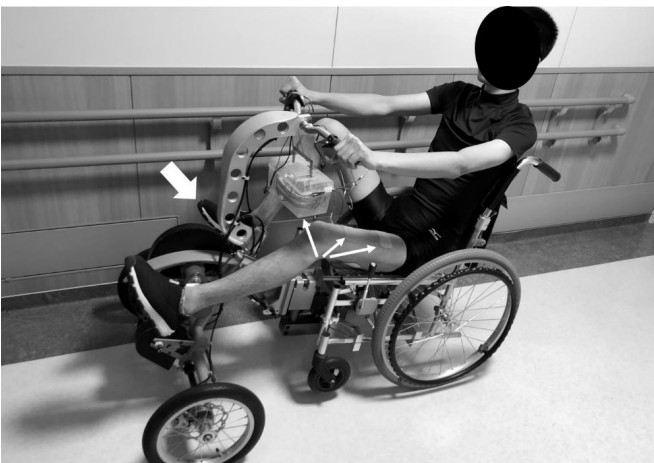

**Figure 3.** A new functional electrical stimulation (FES) cycling system. The FES cycling system consists of a front-wheel drive unit (thick white arrow) that can be attached to a standard wheelchair and an FES control part (thin white arrows) that enables cycling operation for lower limb paralysis.

## 3. Robotic Rehabilitation with FES for SCI

The following factors have been listed as important in planning a rehabilitation program: (1) dose: frequency/long-term repetition of nerve input, (2) quality: quality of nerve input, and (3) paired associative stimulation: synchronization/consistency of nerve input [55]. In gait rehabilitation, the "quantity" of repeating correct gait movements and the "quality" of appropriate neural input according to the gait cycle are considered to be important. The midbrain gait induction field of the brain stem and the central pattern generator (CPG) composed of spinal cord interneurons play a role in the expression and drive of gait movement [56]. It is often difficult to repeat and continue walking training of sufficient quantity and quality via conventional rehabilitation treatment by a physiotherapist alone. With this in mind, rehabilitation robots can provide valuable assistance by providing an amount of training as well as motor learning that cannot be obtained by conventional training.

The pioneer rehabilitation robot for SCI is the Lokomat®, a treadmill walking robot based on the CPG theory of Diez et al. [57]. The Lokomat® is an exoskeleton-based traveling robot that assists stepping by robot control, and achieves walking movement patterns of the lower limbs by motors installed in the hip and knee joints [58]. The effectiveness of walking training by robot rehabilitation for people with SCI has also been reported for Wearable Power-Assist Locomotor (WPAL) and hybrid assistive limb (HAL) [59,60].

It is difficult for paralyzed limbs to perform automatic exercises with muscle contraction even with walking training by robot rehabilitation. However, training can be performed with automatic muscle contraction from the early stage of onset when a robotic system is used together with FES. Muscle fatigue is a known problem with FES alone; however, robot assist can compensate for muscle fatigue, and interactions within the hybrid system prevent disuse, suppress spasticity, and effectively re-educate neuromuscular muscles. We have developed a walking training rehabilitation robot with FES for hemiplegics [61], and the development of equipment for use with paraplegics is underway (Figure 4).

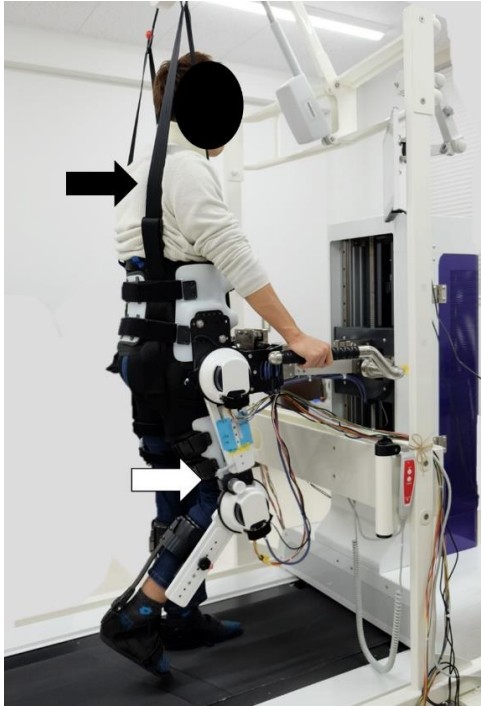

**Figure 4.** A walking training rehabilitation robot with functional electrical stimulation (FES) for paraplegia. The exoskeleton is made based on the long leg orthotic device, and the rehabilitation unloading lift (black thick arrow), FES and robot orthotic device (white thick arrow) are attached, and walking exercise is performed on the treadmill.

## 4. Future Directions of FES for SCI

In recent years, the number of elderly SCI patients has increased [6]. In middle-aged to elderly people, they might have several physical problems such as osteoporosis, muscle atrophy and weakness [62,63]. For treatment of osteoporosis in elderly SCI patients, combination with pharmaceutical treatment and neurorehabilitation has been considered as one of the choices, the meta-analysis by Chang et al. indicates that bisphosphonate administration early following SCI effectively attenuated sublesional bone loss, and FES intervention for chronic SCI patients could significantly increase sublesional bone mineral density near the site of maximal mechanical loading [64]. For prevention and recovery of muscle atrophy in SCI patients, Kern et al. reviewed that atrophic myofibers could be rescued by home-based FES using purpose-developed stimulators and electrodes [63]. Future investigations seem required to reach clinically applicable recommendations of neurorehabilitations for osteoporosis and muscle atrophy as well as weakness in elderly SCI patients.

In addition, a drawback of FES for patients with quadriplegic upper limbs due to cervical SCI is that the person cannot operate the device by themselves. Therefore, the therapist is responsible for activating the stimulation in conventional FES therapy. However, recent technology using brain machine interface (BMI), and advances in BMI technology have enabled direct input into the FES device. Rohm et al. reported recovery of finger function in combination with a BCI and FES for upper limb paralysis due to C4 level cervical SCI [65]. Several recent studies have reported the reconstruction of reach and grip movements of the upper limbs using the BCI and FES systems in cervical SCI with quadriplegia [66,67]. Further developments in rehabilitation using BMI technology and FES are expected in the future.

## 5. Conclusions

In recent years, regenerative medicine for SCI has made great strides, and it is possible that further developments will change SCI treatment completely in the future. FES is expected to play an important role in the regeneration of paralyzed muscles in the field of spinal cord regeneration and rehabilitation medicine. In addition, research and development of rehabilitation robots for paralyzed limbs are progressing dramatically. Rehabilitation robots are highly effective for promoting stability as an assisting force for paralyzed limbs. The combined use of technologies such as regenerative medicine, BMIs, and rehabilitation robot technology with FES is likely to lead to motor reconstruction at the early stage following SCI.

**Author Contributions:** Conceptualization, Y.K. and Y.S.; methodology, D.K., K.S., R.K., S.C. and K.H.; validation, D.K., K.S. and R.K.; formal analysis, D.K. and K.S.; investigation, Y.K.; resources, D.K.; data curation, D.K., K.S., R.K., S.C. and K.H.; writing—original draft preparation, Y.K.; writing—review & editing, D.K., K.S., R.K., S.C., K.H., Y.S. and N.M.; visualization, R.K. and K.H.; supervision, N.M.; project administration, Y.S. All authors have read and agreed to the published version of the manuscript.

**Funding:** This research was funded by JSPS KAKENHI, grant numbers 09671466, 12557122, and 18K10668.

**Institutional Review Board Statement:** Ethical review and approval were waived for this study due to review article.

**Informed Consent Statement:** Informed consent was obtained from all subjects involved in the study. Written informed consent has been obtained from the patient(s) to publish this paper.

**Data Availability Statement:** We did not report any original data in this review article.

**Acknowledgments:** The authors thank for all of the member of Division of Rehabilitation Medicine, Akita University Hospital and Akita Motion Analysis Group.

**Conflicts of Interest:** The authors declare no conflict of interest.

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
