# Peer review of "Advanced Equipment Development and Clinical Application in Neurorehabilitation for Spinal Cord Injury: Historical Perspectives and Future Directions"

_applsci, doi:10.3390/app12094532_

Round 1

Reviewer 1 Report

Major

The paper theme is interesting, however, the manuscript does not reflect the title. The paper is just a simple narrative review without any inclusion/exclusion criteria, with a superficial discussion about the theme.  To a review study, high-quality images are important to improve the understanding, despite that, the manuscript presents only the authors' pictures and does not illustrate the techniques like in C Pasluosta et al 2022 J. Neural Eng. 19 013001 or in https://doi.org/10.1038/s41551-021-00732-x. 

Minor

L. 11: "Spinal cord injury (SCI) is becoming increasingly common in rapidly aging societies such as Japan", and the other countries (mainly in development)? Your work title does not involve just Japan. Include similar to begin in the introduction or change the phrase.

L. 13 and L. 59: change "nerve plasticity" to "neural plasticity". The nerve is related to the peripherical nervous system. The spinal cord belongs to the central nervous system (except cauda equina).

Author Response

Responses to the Reviewers’ Comments (Ms. ID applsci-1647252)

The comments from the academic editor and reviewers have helped us to improve the quality of our manuscript. We would like to thank the reviewers for their time and effort in reviewing the manuscript.

Reviewer 1.

Major

The paper theme is interesting, however, the manuscript does not reflect the title. The paper is just a simple narrative review without any inclusion/exclusion criteria, with a superficial discussion about the theme. To a review study, high-quality images are important to improve the understanding, despite that, the manuscript presents only the authors' pictures and does not illustrate the techniques like in C Pasluosta et al 2022 J. Neural Eng. 19 013001 or in https://doi.org/10.1038/s41551-021-00732-x.

Thank you for the valuable comment. We have illustrated the schematic diagram of the principle of functional electrical stimulation (FES) as Figure 1. The figure explains that instead of upper motor neuron damaged by spinal cord injury, muscles contract and reconstruct function through lower motor neurons that are not damaged by controlled electrical stimulation. Since Figure 1 has been added, Figures 1 to 4 of the first draft have been revised to Figures 2 to 5, respectively.

Minor

  1. 11: "Spinal cord injury (SCI) is becoming increasingly common in rapidly aging societies such as Japan", and the other countries (mainly in development)? Your work title does not involve just Japan. Include similar to begin in the introduction or change the phrase.

Thank you for the comment. This review is not focus only in Japan as the comment from reviewer, so we have deleted the first phrase of the abstract.

  1. 13 and L. 59: change "nerve plasticity" to "neural plasticity". The nerve is related to the peripherical nervous system. The spinal cord belongs to the central nervous system (except cauda equina).

Thank you for the comment. We have edited the “nerve plasticity” to “neural plasticity” in the revised manuscript.

Reviewer 2 Report

The article describes the history of FES in rehabilitation after SCI. In the introduction they refer to the increasing number of older patients with SCI after falls. In their citations however, only younger patients were included in these studies with low patient numbers. Even for these patients no routine use has yet been achieved. For older people other problems will occur due to muscular atrophy, sarkopenia, visual impairment and/or osteoporosis; no mention is made of these problems in the review.

Although there is a large need for neurorehabilitation devices, especially for older patients, the manuscript does not address the peculiarities of older patients. This should be included in the manuscript since the authors emphasize this aspect in their introduction.

Abstract: the authors mention „brain machine interface“ but do not add any future direction in this line nor do they mention the research in this area. I am no expert in this area; to my knowledge some prototypes are already in testing (Bockbrader et al., PM&R, 2018, 10(9Suppll.- 2) S233-S243; Sebastian-Romagosa et a., Front. Neurosci. 2020, doi.org/10.3389/fnins.2020.591435)

The last chapter „robotic rehabilitation for SCI“ covers an entirely different system, i.e. not nerve stimulation, but muscular support by an active exoskeleton. In my view this should not be included in a „functional electrical stimulation“ review.

Specifically, in a review I would expect a decription of the value of FES as compared to other methods. All studies referred to ar uncontrolled studies, which makes a comparison difficult but the authors should at least address this point (they only do it on page two in a cursory way).

Author Response

Responses to the Reviewers’ Comments (Ms. ID applsci-1647252)

The comments from the academic editor and reviewers have helped us to improve the quality of our manuscript. We would like to thank the reviewers for their time and effort in reviewing the manuscript.

Reviewer 2

The article describes the history of FES in rehabilitation after SCI. In the introduction they refer to the increasing number of older patients with SCI after falls. In their citations however, only younger patients were included in these studies with low patient numbers. Even for these patients no routine use has yet been achieved. For older people other problems will occur due to muscular atrophy, sarkopenia, visual impairment and/or osteoporosis; no mention is made of these problems in the review.

Thank you for the valuable comment. It has been reported that the physical function and activity of daily living (ADL) of older patients with SCI do not improve significantly [7. Wirz M et al. Neurotrauma 2015]. In addition, in the recent report, one-month mortality risk was significantly higher at older than 75 years compared to younger than 55 years in the patients with traumatic SCI [9. Barbiellini Amidei C et al. Spinal Cord. 2022]. We have mentioned these points in the Introduction with references. We also have mentioned the several physical dysfunction of elderly SCI patients in the Discussion of future directions of FES paragraph. The middles-aged to elderly patients with SCI have several physical problems such as osteoporosis and sarcopenia [63. Haider I.T. Osteoporos Int. 2018, 64. Kern H et al. Neurol Res. 2017]. We have reviewed several papers regarding these problems and mentioned it in the future diresctions of FES for SCI paragraph.

Although there is a large need for neurorehabilitation devices, especially for older patients, the manuscript does not address the peculiarities of older patients. This should be included in the manuscript since the authors emphasize this aspect in their introduction.

Thank you for the comment. Could you please refer the response to the above comment? We have described this point in the 4) Future directions of FES for SCI paragraph. In middle-aged to elderly people, they might have several physical problems such as os-teoporosis, sarcopenia which reveals muscle atrophy and weakness [63. Haider I.T. Osteoporos Int. 2018, 64. Kern H et al. Neurol Res. 2017]. For treatment of osteoporosis in elderly SCI patients, combination with pharmaceutical treatment and neurorehabilitation have been considered as one of the choices, the meta-analysis by Chang et al. indicates that bisphosphonate administration early following SCI effectively attenuated sublesional bone loss, and FES intervention for chronic SCI patients could significantly increase sublesional bone mineral density near the site of maximal mechanical loading [65 Chang KV et al. 2013 PLos One]. For prevention and recovery of muscle atrophy in SCI patients, Kern et al. reviewed that atrophic myofibers could be rescued by home-based FES using purpose developed stimu-lators and electrodes [66. Kern H et al. 2017, Neurol Res.]. Future investigations seem required to reach to clinically applicable recommendations of neurorehabilitations for os-teoporosis and muscle atrophy as well as weakness in elderly SCI patients.

Abstract: the authors mention „brain machine interface“ but do not add any future direction in this line nor do they mention the research in this area. I am no expert in this area; to my knowledge some prototypes are already in testing (Bockbrader et al., PM&R, 2018, 10(9Suppll.- 2) S233-S243; Sebastian-Romagosa et a., Front. Neurosci. 2020, doi.org/10.3389/fnins.2020.591435)

Thank you for the important comment. As the comment from reviewer, a brain-machine interface or brain computer interface (BCI) should be one of an important future direction of FES therapy for SCI. We have mentioned this point at the new paragraph, which is 4) Future directions of FES, in the revised manuscript.

In addition, a drawback of FES for patients with quadriplegic upper limbs due to cervical SCI is that the person cannot operate the device by themselves. Therefore, the therapist is responsible for activating the stimulation in conventional FES therapy. However, recent technology using brain machine interface (BMI), Advances in BMI technology have enabled direct input into the FES device. Rohm et al. re-ported recovery of finger function in combination with a BCI and FES for upper limb pa-ralysis due to C4 level cervical SCI [67 Rohm M. et al Artif. Intell. Med. 2013]. Several recent studies have reported the reconstruction of reach and grip movements of the upper limbs using the BCI and FES systems in cervical SCI with quadriplegia [68 Ajiboye A.B et al. Lancet 2017, 69 Jovanovic LI, et al. 2021 Spinal Cord Ser Cases.]. Further developments in rehabilitation using BMI technology and FES are expected in the future.

The last chapter „robotic rehabilitation for SCI“ covers an entirely different system, i.e. not nerve stimulation, but muscular support by an active exoskeleton. In my view this should not be included in a „functional electrical stimulation“ review.

Thank you for the important comment. In our institute, we have developed the robotic rehabilitation machine with FES for SCI patients. Although, an exoskeleton-based traveling robot such as Lokomat shows effectiveness of walking training, it is difficult for paralyzed limbs to perform automatic exercise with muscle contraction with exoskeleton-based robot. We have considered that an automatic muscle contraction from the early stage by robot rehabilitation, thus we have developed the walking training rehabilitation robot with FES. Therefore, we would like to include this walking training rehabilitation robot with FES in this review. Could please understand the meaning of this walking training rehabilitation robot with FES.

Specifically, in a review I would expect a decription of the value of FES as compared to other methods. All studies referred to ar uncontrolled studies, which makes a comparison difficult but the authors should at least address this point (they only do it on page two in a cursory way).

Thank you for the comment. The randomized controlled trials performed to investigate the effectiveness of FES training on the independence or function of SCI patients. Popovic et al. performed the randomized controlled trial to examine the efficacy of 40 hours of FES therapy with conventional occupational therapy (COT) compared with COT alone [24. Popovic M et al. Neurorehabil. Neural Repair 2011]. The FES therapy significantly reduced disability and improved voluntary grasping in the subjects with tetraplegia. Kapadia et al. reported that 16 weeks of thrice-weekly FES-assisted walking program compared to non-FES exercise program in chronic incomplete traumatic SCI patients [25. Kapadia et al. J Spinal Cord Med. 2014]. In the recent systematic review and meta-analysis has indicated that FES although significantly increased upper extrem-ity independence, there was no obvious difference in overall upper extremity function, lower extremity independence, and life quality of individuals with SCI [26. Duan R et al. Spine 46, E398-410, 2021]. We have described these description of the value of FES in the last paragraph of 2. FES for SCI in the revised manuscript.

Round 2

Reviewer 1 Report

Ok.